# Ensemble Mashups: A Simple Recipe For Better Bayesian Optimization

**Anand Ravishankar, Fernando Llorente, Yuanqing Song, Petar Djuric**
Department of Electrical and Computer Engineering
Stony Brook University

## Abstract

Bayesian optimization (BO) is a popular approach to optimizing costly, black-box functions that rely on a statistical surrogate model of the function, typically a Gaussian process (GP) and the so-called acquisition function (AF). Although the choices of the GP kernel and the AF can strongly affect the results, there does not exist an automatic way of selecting them. Ensembling, namely, using several, different kernels (Multi-Model) or AFs (Multi-AF) is one possibility for deriving a BO algorithm that is robust and safer. These ideas have been considered separately in the past. In this work, we consider ensembles of both kernels and AFs (Multi-Model-Multi-AF) and perform an empirical comparison to show their superiority with respect to single-ensemble algorithms.

## 1 Introduction

Bayesian optimization (BO) has become an important tool for black-box optimization and is widely used in fields such as machine learning, automated design, and complex system optimization [1; 2; 3]. It predicts the behavior of the objective function by positing a proxy model, usually a Gaussian process (GP) [4] and uses these predictions to guide the optimization process, greatly improving the efficiency of the algorithm. However, in BO, the choice of kernel function and acquisition function (AF) is crucial to the final optimization result. The choice of the kernel affects the smoothness and complexity of the model. The AF guides how to select new points for evaluation. If the AF cannot effectively explore-exploit the search space, it may fall into a local optimum or waste computing resources. Different choices will directly determine the performance and efficacy of the optimization process. Therefore, how to select and optimize these components has become an important issue in current research [2; 1].

Ensembling is a popular strategy in machine learning for leveraging multiple learning algorithms and improving the performance [5]. Indeed, the advantages of ensembling in BO are well-recognized, appearing in the majority of the best-performing algorithms that took part in the recent competition [6]. Ensembling is part of a recent federated BO [7], which can be interpreted as using a mixture of GP models. Ensemble models have also gained popularity in transfer learning extensions to BO, where information from base tasks are combined via weighted rankings of individual predictions or via weighted combination of AFs [8; 9; 10]. Still, as noted in [6], more studies are needed to understand ensembling in BO.

In this work, we focus on ensembles of two important components in GP-based BO: kernels and AFs. The ensembling strategy that we consider consists of sampling one representative from a pool of candidates. Existing works have been limited to either different kernel functions or different AFs. For example, the GP-hedge algorithm [11] improves optimization performance by integrating several acquisition functions, but still uses a single pre-selected kernel. This algorithm is an instance of a Multi-AF (MA) method. At the same time, Bayesian Model Averaging (BMA)

Workshop on Bayesian Decision-making and Uncertainty, 38th Conference on Neural Information Processing Systems (NeurIPS 2024).

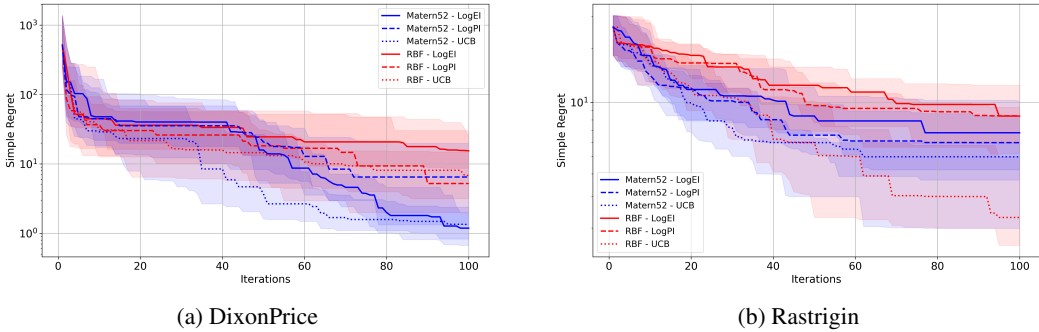

Figure 1: Simple regret in log-scale. The lines represents the median results across 25 simulations, with the bands representing the 0.75 and 0.25 quantiles.

improves optimization results by integrating different kernel functions [12], but does not adopt an ensembling strategy for AFs. This algorithm is an instance of a Multi-Model (MM) method. Although these methods have proved to be useful, their interaction has not been explored in the literature. To address these issues, we propose an MMMA (Multi-Model-Multi-AF) approach to BO. This approach fills the gaps in existing methods by integrating kernel functions and AFs at the same time. We conducted experiments on 16 benchmarks and showed empirical evidence that MMMA-BO provides a more robust approach to BO.

**Related Work.** Some previous works that have considered using ensembles of AFs are [2; 11; 13; 14; 15; 16; 17]. "Ensembling" is a very broad term that can interpreted as (i) individually optimizing a portfolio of AFs and choosing one of the maximizers according to some criteria [11; 13]; (ii) selecting and optimizing one AF from the portfolio [14; 15]; (iii) jointly optimizing several AFs in a multi-objective fashion [16; 17]; or (iv) integrating out the AF using an approximate fully Bayesian GP [2]. The use of an ensemble of models (kernels) can be also interpreted differently [2; 12; 18].

The fully Bayesian treatment of GP hyperparameters naturally leads to an ensemble of kernels, each one with a hyperparameter sample, drawn approximately from the posterior; this model in conjunction with, e.g., the expected improvement (EI) AF, leads to the "integrated" AF mentioned above [2]. On the other hand, a series of different kernels can be weighted according to their marginal likelihoods (i.e., the probability of the data given the model/kernel) using BMA weights; then, at each iteration, a kernel is selected according to its BMA weight [12; 18].

## 2 Multi-Model Multi-Acquisition function Bayesian optimization

**Background.** We aim to solve $\max_{\mathbf{x}} f(\mathbf{x})$ by sequentially acquiring a dataset $\mathcal{D}_t = \{\mathbf{X}_t, \mathbf{y}_t\} = \{(\mathbf{x}_i, y_i)\}_{i=1}^t$, where $y_i = f(\mathbf{x}_i) + u_i$, $u_i \sim N(0, \sigma^2)$. The function $f(\mathbf{x})$ is assumed to be sampled from a GP, $f(\mathbf{x}) \sim \mathcal{GP}(\mu(\mathbf{x}), \kappa(\mathbf{x}, \mathbf{x}'))$, where $\mu(\mathbf{x}) = 0$ and $\kappa(\mathbf{x}, \mathbf{x}')$ is the kernel function. At each iteration, the function is queried at $\mathbf{x}_{t+1} = \arg\max_{\mathbf{x}} \alpha(\mathbf{x}; \mathcal{D}_t)$, collecting $y_{t+1} = f(\mathbf{x}_{t+1}) + u_t$ and updating $\mathcal{D}_{t+1} = \mathcal{D}_t \cup \{(\mathbf{x}_t, y_t)\}$. The criterion $\alpha(\mathbf{x}; \mathcal{D}_t)$ is the AF which is computed using the posterior GP, $f|\mathcal{D}_t \sim \mathcal{GP}(m_t(\mathbf{x}), s_t^2(\mathbf{x}))$, where $m_t(\mathbf{x})$ and $s_t^2(\mathbf{x})$ are the posterior predictive mean and variance of $f(\mathbf{x})$ at $\mathbf{x}$ [4]. Although the particular choices of $\kappa$ and $\alpha$ determine the performance of the algorithm, they are usually kept fixed throughout the algorithm.

**Motivation.** Consider a scenario where we want to implement a BO algorithm but have limited prior knowledge about the target function and insufficient expertise in selecting a suitable AF. The standard recommendations for the kernel $\kappa$ and AF $\alpha$ are often the Matérn 5/2 kernel and Expected Improvement (EI), respectively. However, these defaults may not be "optimal" for our specific problem. Ideally, if we could experiment with different combinations of $(\kappa, \alpha)$, we would identify the best pair in hindsight. Furthermore, when applying BO across various problems, different combinations may yield better results. Unless one combination is clearly dominant, selecting the "best" pair remains challenging—even in hindsight. For example, Figure 1 illustrates the regret for several combinations of $\kappa \in \{\texttt{Matérn-5/2}, \texttt{RBF}\}$ and $\alpha \in \{\texttt{EI, PI, UCB}\}$ for two different test functions. For either function, the simple regret after 100 iterations varies by roughly one order of magnitude

depending on the choice of $(\kappa, \alpha)$. Notably, the best-performing configuration for the Rastrigin function, (RBF, UCB), is not among the top two configurations for the DixonPrice function.

**MMMA-BO.** Let us consider $M$ different kernels $\{\kappa_m\}_{m=1}^M$ and $N$ different AFs $\{\alpha_n\}_{n=1}^N$. Our goal is to ensemble both kernels and AFs using two sets of normalized weights $\{w_m^\kappa\}_{m=1}^M$ and $\{w_n^\alpha\}_{n=1}^N$ such that $w_m^\kappa \geq 0$ and $w_n^\alpha \geq 0$ represent respectively the probability of choosing kernel $\kappa_m$ and AF $\alpha_n$ at each iteration. By using multiple kernels and AFs simultaneously, we avoid making a potentially suboptimal choice at the outset. The MMMA-BO algorithm in 1 is essentially the same as standard BO but it involves sampling a pair—one kernel and one AF—at each iteration, rather than committing to a single combination throughout the optimization process.

The sampling probabilities for each kernel and AF can either be fixed in advance or adaptively adjusted based on performance during the optimization process. A straightforward and effective fixed choice is to assign uniform weights, $w_m^\kappa = \frac{1}{M}$ and $w_n^\alpha = \frac{1}{N}$, corresponding to the random selection of a different BO algorithm at each iteration. However, a performance-guided approach offers a more dynamic alternative, where "rewards" are assigned to kernels and AFs based on their contribution to the optimization process. For kernels, setting the weights with BMA [12], $w_m^\kappa \propto p(\mathbf{y}_t|\kappa_m)$, where $p(\mathbf{y}_t|\kappa_m)$ is the GP marginal likelihood with optimized hyperparameters, makes sense from a theoretical perspective. This method allows for kernel selection based on their fit to the observed data. Nevertheless, in low-data regimes—typical in BO—this approach may not always be appropriate. For AFs, the $w_n^\alpha$ can be adjusted based on rewards reflecting each $\alpha_n$'s contribution to the optimization. For instance, we reward $\alpha_n$ if it identifies a candidate with a function value greater than the current best [15]. A particular case of this strategy is the Hedge algorithm [11] ("Bandit" in the experiments).

---

**Algorithm 1** Bayesian Optimization with Multiple Kernels and Acquisition Functions

---

1: **Input:** Set of kernels $\{\kappa_m\}_{m=1}^M$, set of acquisition functions $\{\alpha_n\}_{n=1}^N$, initial dataset $\mathcal{D}_0$,
2: Kernel weights $\{w_m^\kappa\}_{m=1}^M$, acquisition function weights $\{w_n^\alpha\}_{n=1}^N$
3: **Initialize:** $\mathcal{D} = \mathcal{D}_0$
4: **for** $t = 1$ to $T$ **do**
5:     (i) Sample a kernel $\kappa^*$ from $\{\kappa_m\}_{m=1}^M$ using the weights $\{w_m^\kappa\}_{m=1}^M$
6:     (ii) Fit the Gaussian Process (GP) with kernel $\kappa^*$ to the dataset $\mathcal{D}$
7:     (iii) Sample an acquisition function $\alpha^*$ from $\{\alpha_n\}_{n=1}^N$ using the weights $\{w_n^\alpha\}_{n=1}^N$
8:     (iv) Optimize the acquisition function $\alpha^*$ to find the next query point $\mathbf{x}_{t+1}$
9:     (v) Evaluate $y_{t+1} = f(\mathbf{x}_{t+1}) + u_{t+1}$ and update the dataset: $\mathcal{D} = \mathcal{D} \cup \{(\mathbf{x}_{t+1}, y_{t+1})\}$
10:     **if** weight_update **then**
11:         (vi) Update the kernel weights $\{w_m^\kappa\}_{m=1}^M$ and/or acquisition function weights $\{w_n^\alpha\}_{n=1}^N$
12:     **end if**
13: **end for**
14: **Return:** The best found point $\mathbf{x}_{\text{best}}$

---

## 3 Experiments

The importance of ensembling can be demonstrated by contrasting MMMA against algorithms of varying degrees of ensembling at different levels (kernel or AF) and different strategies of ensembling. In this study, we considered a pool of kernels including RBF, Matérn 5/2, and Matérn 3/2, along with the set of AFs, EI, PI and UCB. These design choices were guided by their popularity and general applicability. For more details, see Table 2 in the appendix. The algorithms were evaluated on 16 test functions from Botorch's suite [19] (see Table 1 in the appendix). We ran 25 independent simulations where, at each simulation, we ran BO for 100 iterations.

We consider 3 metrics to evaluate the algorithms: (1) the gap metric, (2) simple regret, and (3) cumulative regret. Each metric focuses on a different aspect of the optimizer's behavior. The gap metric measures improvement relative to the starting condition, simple regret quantifies the distance between the best-observed value and the global optimum at each iteration, and cumulative regret quantifies the rate at which regret accumulates over time.

**The gap metric.** Figure 2-(a) presents boxplots of the final gap metric, illustrating that increased ensembling leads to better results. This improvement is due to the optimizer having access to a

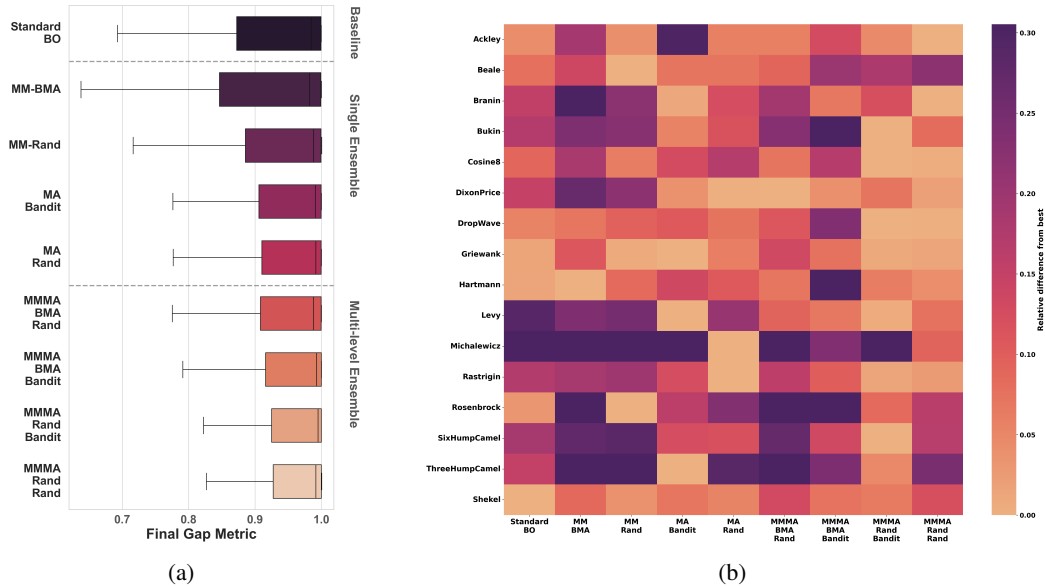

|     |     |
| :-: | :-: |
| (a) | (b) |

Figure 2: (a) Final gap metric accumulated across all test functions. (b) Heat map of relative differences in the order of polynomial. The methods are categorized into three groups: Baseline (Standard BO), Single Ensemble (MultiModel, MultiAF), and Multi-level Ensemble (MMMA variants).

broader selection of kernels and AFs. Notably, the baseline performs similarly to kernel-level ensembling, which can be attributed to the similarity among the kernels used in our study. However, when ensembling is extended to include AFs, there is a significant improvement due to the greater diversity among the AFs. Importantly, the results of MMMA are more concentrated to the right, indicating less variability and greater robustness in performance compared to other methods.

**Regret growth.** Given a function class $F$, a common objective is to find an optimizer $\pi$ which achieves sublinear rate of regret for all functions $f \in F$. Formally, $\forall f \in F$, the objective is to have $\lim_{t \to \infty} \frac{R_t(\pi, f)}{t} = 0$, where $R_t$ is the cumulative regret at iteration $t$. We can make this objective stricter by imposing a functional form on the rate of regret. For example, $\forall f \in F$, $R_t(\pi, f) \leq Ct^b$, $C > 0$ and $b < 1$. A good optimizer has $b \ll 1$. To determine the value of $b$, we fit a polynomial to the median cumulative regret using non-linear least squares. Figure 2-(b) shows the relative difference in $b$ values of each optimizer $\pi_i$ to the best optimizer $\pi^*$ for every function, namely, the lightest cell is the optimizer with the smallest $b$. We can see that MMMA (especially random-bandit and random-random) usually have the lowest $b$ values or they are quite close to the best optimizer. It is again interesting to note that ensembling at the AF level yields significant improvement, while ensembling at the kernel level provides marginal improvement.

**Random v/s Adaptive weights.** We observe that random weights (in both AF and kernel space) tend to perform better than adaptive weight selection. When the adaptive strategy is employed, the "best" performing AF/ kernel will dominate the other candidates and cause the ensemble to collapse to a single choice (see Figs. 3-4). Indeed, with these strategies, the ensemble collapses to an exploitative choice: in BMA, the rewards are assigned to kernels that fit better the current data; in bandit, an AF that suggests points with large function values will be preferred. Hence, these strategies do not reward exploration.

# 4   Conclusions

In this work, we described MMMA-BO, a framework for combining multiple kernels and acquisition functions in BO. The MMMA-BO algorithm involves sampling one kernel and one acquisition function at each iteration, rather than committing to a single combination throughout the optimization process. We conducted experiments on 16 benchmark functions and systematically compared them with existing baseline methods. The experimental results confirm that the MMMA approach pro-

vides more robust performance. In future work, we aim to explore better procedures for determining the ensemble weights and deriving theoretical bounds on the cumulative regret of MMMA-BO.

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

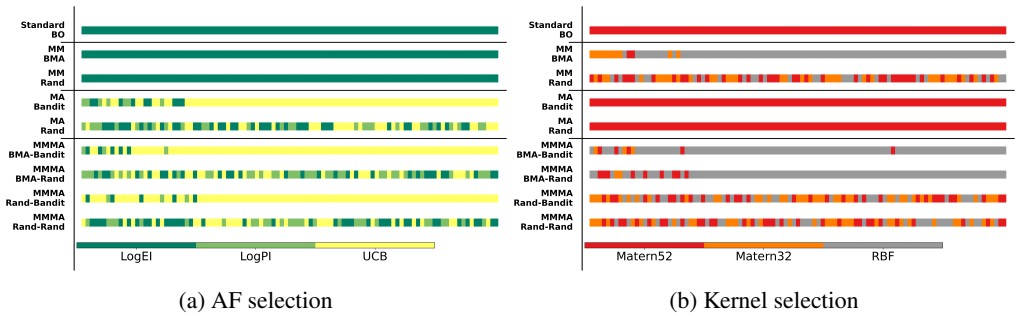

(a) AF selection

(b) Kernel selection

Figure 3: AF and kernel selection for different ensemble levels and strategies for the Ackley function

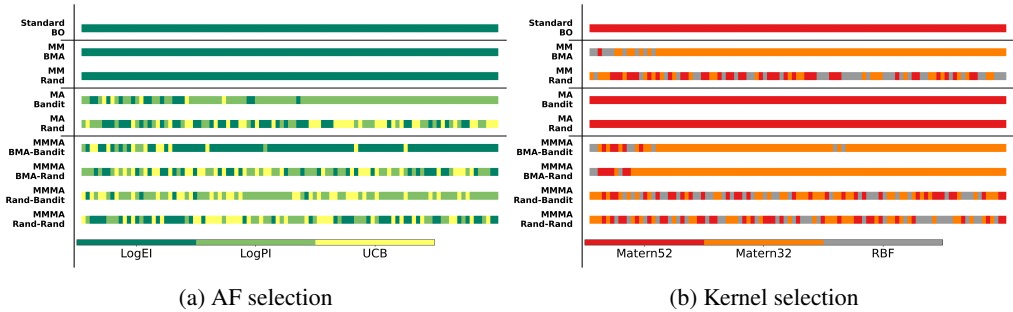

(a) AF selection

(b) Kernel selection

Figure 4: AF and kernel selection for different ensemble levels and strategies for the Hartmann function

## A    Additional details of numerical experiments

All the experiments were conducted using Botorch's open-sourced codebase [19] on an Apple M2 chip with 10 CPU cores and 16GB of memory. The code is hosted at this Github repository: Github. Instructions regarding the code execution can be found in a README file inside the directory "./MMMA". In each algorithm, we use type-II maximum likelihood to fit the kernel hyperparameters. For the optimization of the AF, we use 20 raw samples and 2 random restarts. All the kernels have automatic relevance determination (ARD). For estimating the degree of the polynomial fitted to the median cumulative regret we use the Levenberg–Marquardt algorithm. Plots for cumulative regret, simple regret and gap metric for all 16 functions are shown in Figures 5-8.

## B    Limitations

Design choices like AF and kernel candidates, the weighing parameter of UCB, number of restarts or raw samples of the optimizer, the test functions, are standard limitations of a BO problem. Additionally, as noted by [11], when employing a bandit (or BMA) strategy, the selection often converges eventually to an exploitative strategy. This convergence behaviour can be seen in Figures 3 and 4. There might be scenarios where converging to exploitative strategies is not beneficial, especially if majority of the function's landscape is not explored and the optimizer is stuck in a local minima. Furthermore, the current MMMA setup only support single-objective test functions and analytical AFs. Future work will primarily be geared towards exploring better selection strategies and degree of ensembling. We will also include a larger array of test functions, support for multi-objective test functions and Monte-Carlo base AFs. Providing theoretical bounds for the gap metric and rate of regret will also be explored.

| Function | Dimension | Function | Dimension |
|----------|-----------|----------|-----------|
| Ackley | 4 | Hartmann | 6 |
| Beale | 2 | Levy | 3 |
| Branin | 2 | Michalewicz | 2 |
| Bukin | 2 | Rastrigin | 3 |
| Cosine8 | 8 | Rosenbrock | 2 |
| DixonPrice | 3 | SixHumpCamel | 2 |
| DropWave | 2 | ThreeHumpCamel | 2 |
| Griewank | 5 | Shekel | 4 |

Table 1: Optimization Test Functions with Equations

| Ensemble Level | Method | Ensemble Strategy | Kernel Pool | AF Pool |
|----------------|--------|-------------------|-------------|---------|
| No Ensemble | Standard BO | Traditional Bayesian Optimization | Matern52 | LogEI |
| Single-level Ensemble | MM-BMA | Multi-model with BMA | [RBF, Matern52, Matern32] | LogEI |
| | MM-Rand | Multi-model with random model selection | [RBF, Matern52, Matern32] | LogEI |
| | MA-Bandit | GP-Hedge with bandit-based AF selection | Matern52 | [LogPI, LogEI, UCB] |
| | MA-Rand | GP-Hedge with random AF selection | Matern52 | [LogPI, LogEI, UCB] |
| Two-level Ensemble | MMMA-BMA-Rand | Multi-model BMA, multi-acquisition with random AF selection | [RBF, Matern52, Matern32] | [LogPI, LogEI, UCB] |
| | MMMA-BMA-B | Multi-model BMA, multi-acquisition with bandit-based AF selection | [RBF, Matern52, Matern32] | [LogPI, LogEI, UCB] |
| | MMMA-Rand-B | Multi-model with random model selection, multi-acquisition with bandit-based AF selection | [RBF, Matern52, Matern32] | [LogPI, LogEI, UCB] |
| | MMMA-Rand-Rand | Multi-model with random model selection, multi-acquisition with random AF selection | [RBF, Matern52, Matern32] | [LogPI, LogEI, UCB] |

Table 2: Description of Bayesian Optimization Methods Grouped by Ensemble Levels

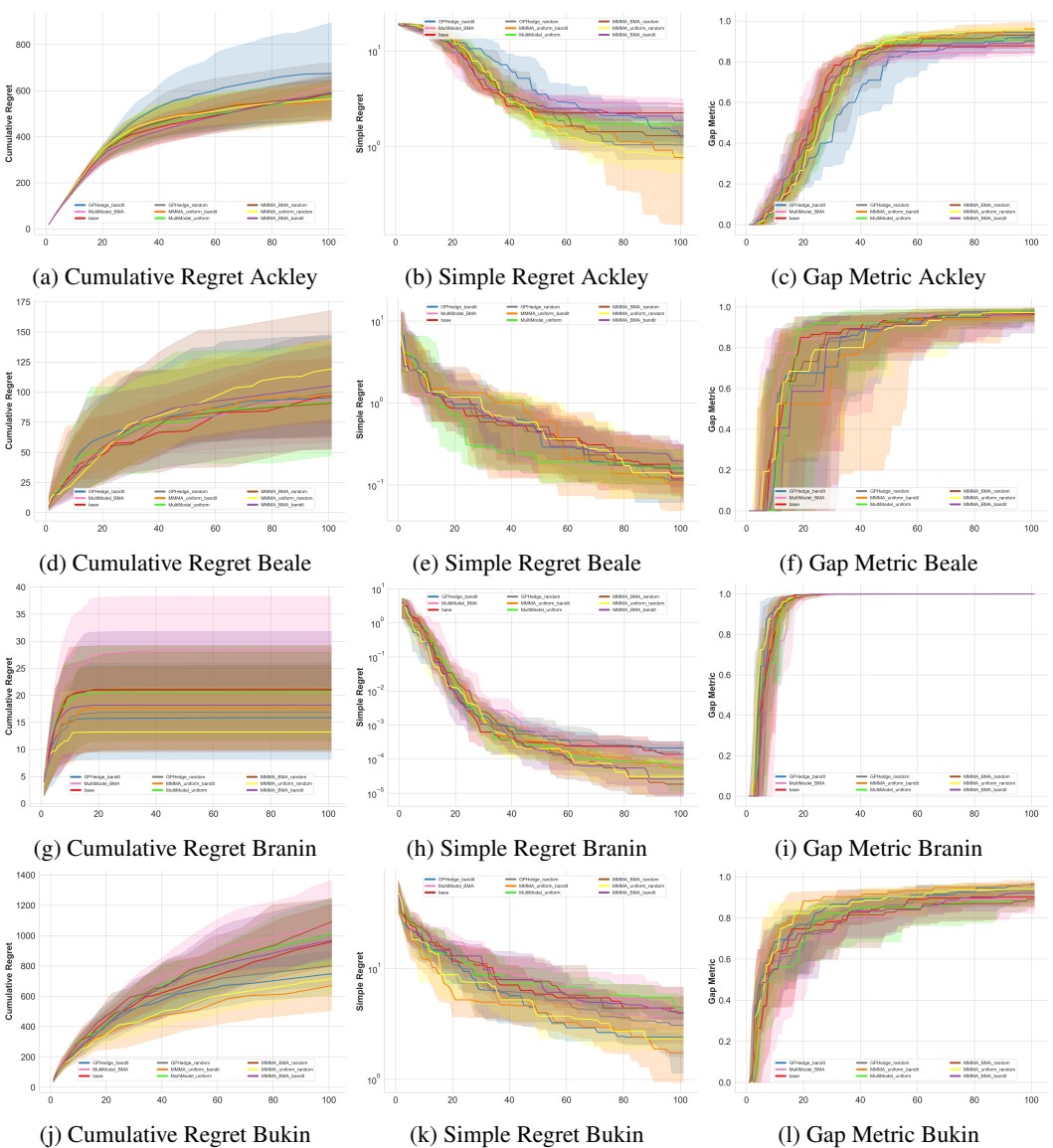

Figure 5: Cumulative Regret, Simple Regret and Gap metric

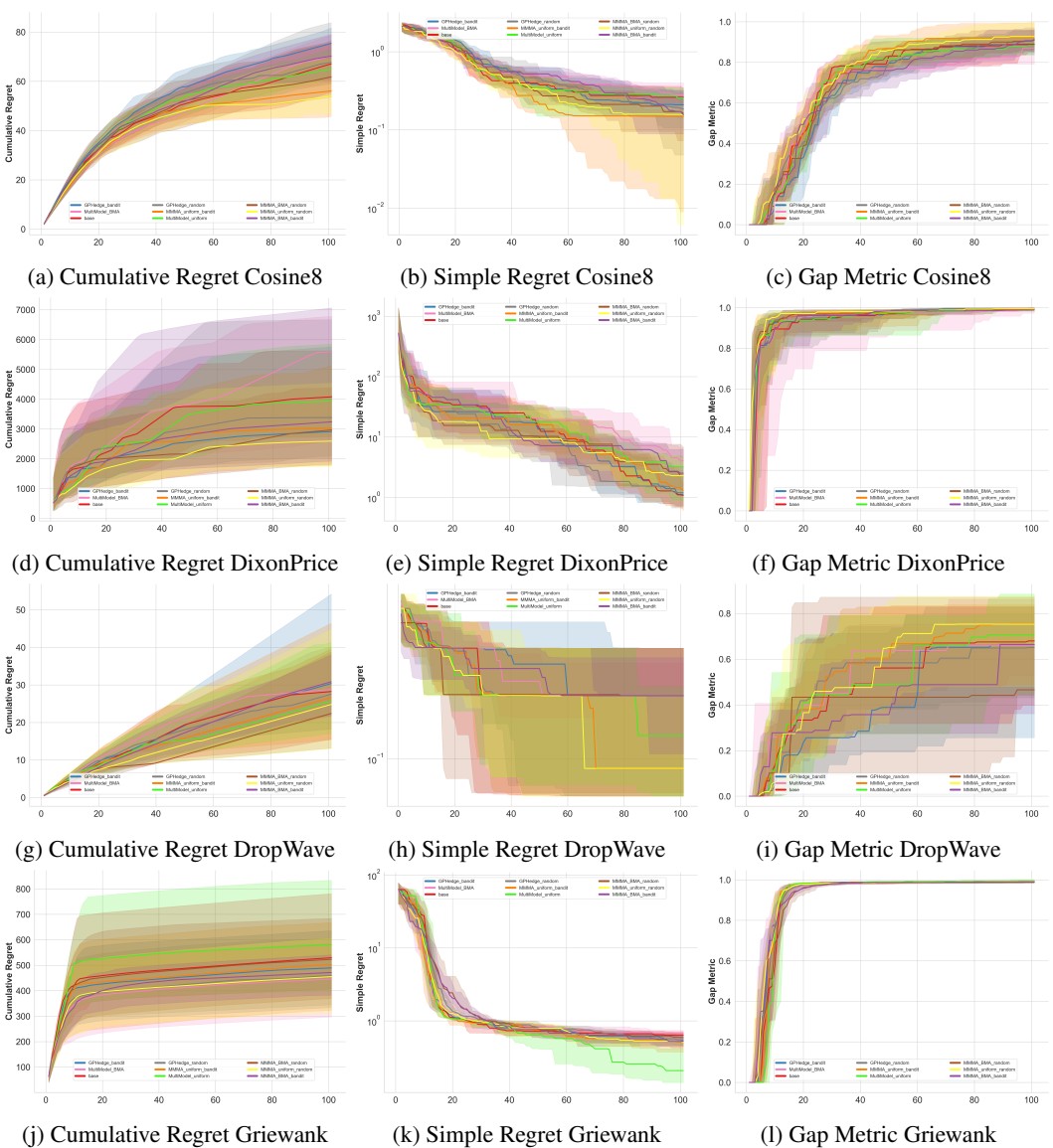

(a) Cumulative Regret Cosine8  (b) Simple Regret Cosine8  (c) Gap Metric Cosine8

(d) Cumulative Regret DixonPrice  (e) Simple Regret DixonPrice  (f) Gap Metric DixonPrice

(g) Cumulative Regret DropWave  (h) Simple Regret DropWave  (i) Gap Metric DropWave

(j) Cumulative Regret Griewank  (k) Simple Regret Griewank  (l) Gap Metric Griewank

Figure 6: Cumulative Regret, Simple Regret and Gap metric

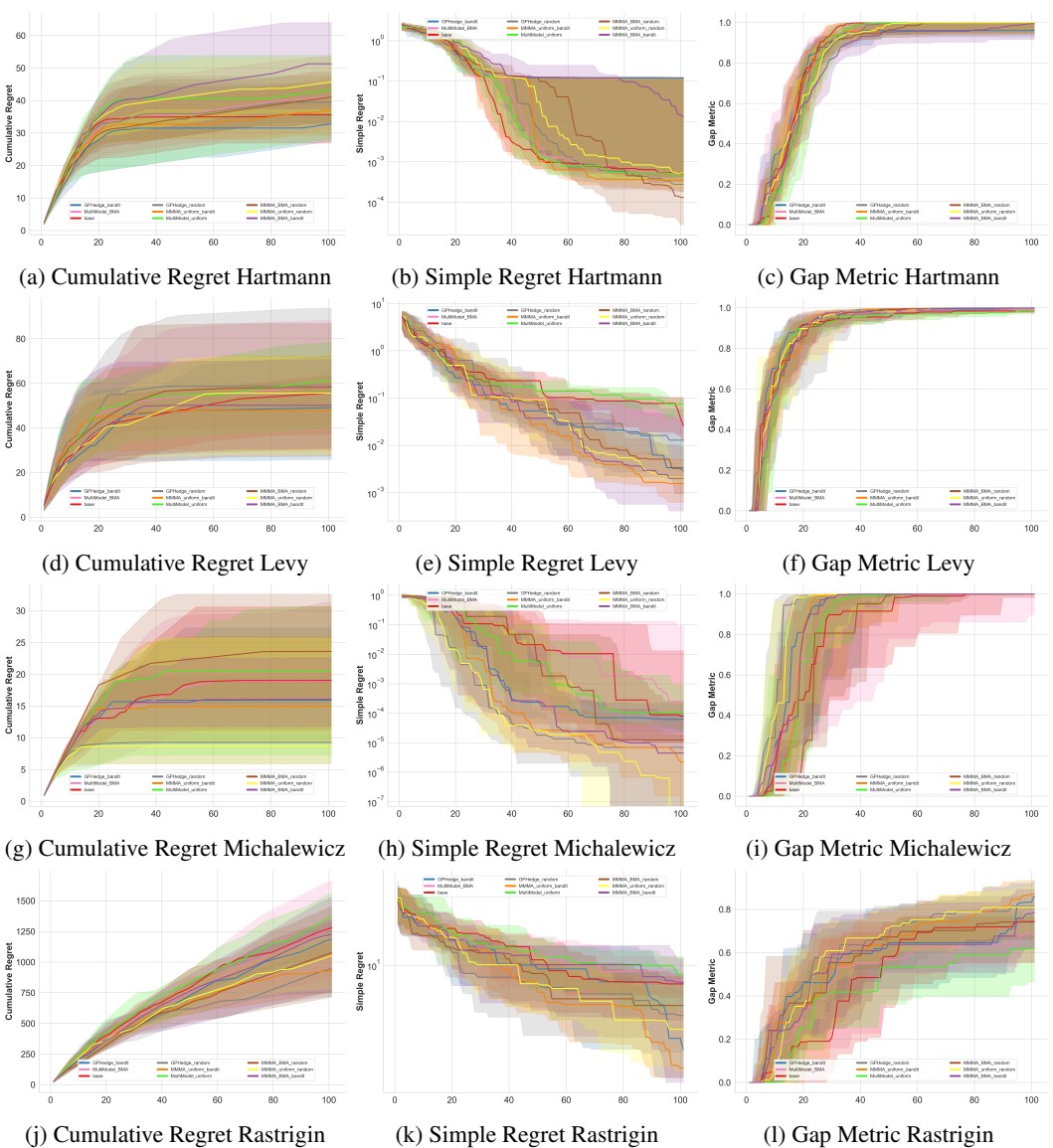

(a) Cumulative Regret Hartmann     (b) Simple Regret Hartmann     (c) Gap Metric Hartmann

(d) Cumulative Regret Levy     (e) Simple Regret Levy     (f) Gap Metric Levy

(g) Cumulative Regret Michalewicz     (h) Simple Regret Michalewicz     (i) Gap Metric Michalewicz

(j) Cumulative Regret Rastrigin     (k) Simple Regret Rastrigin     (l) Gap Metric Rastrigin

Figure 7: Cumulative Regret, Simple Regret and Gap metric

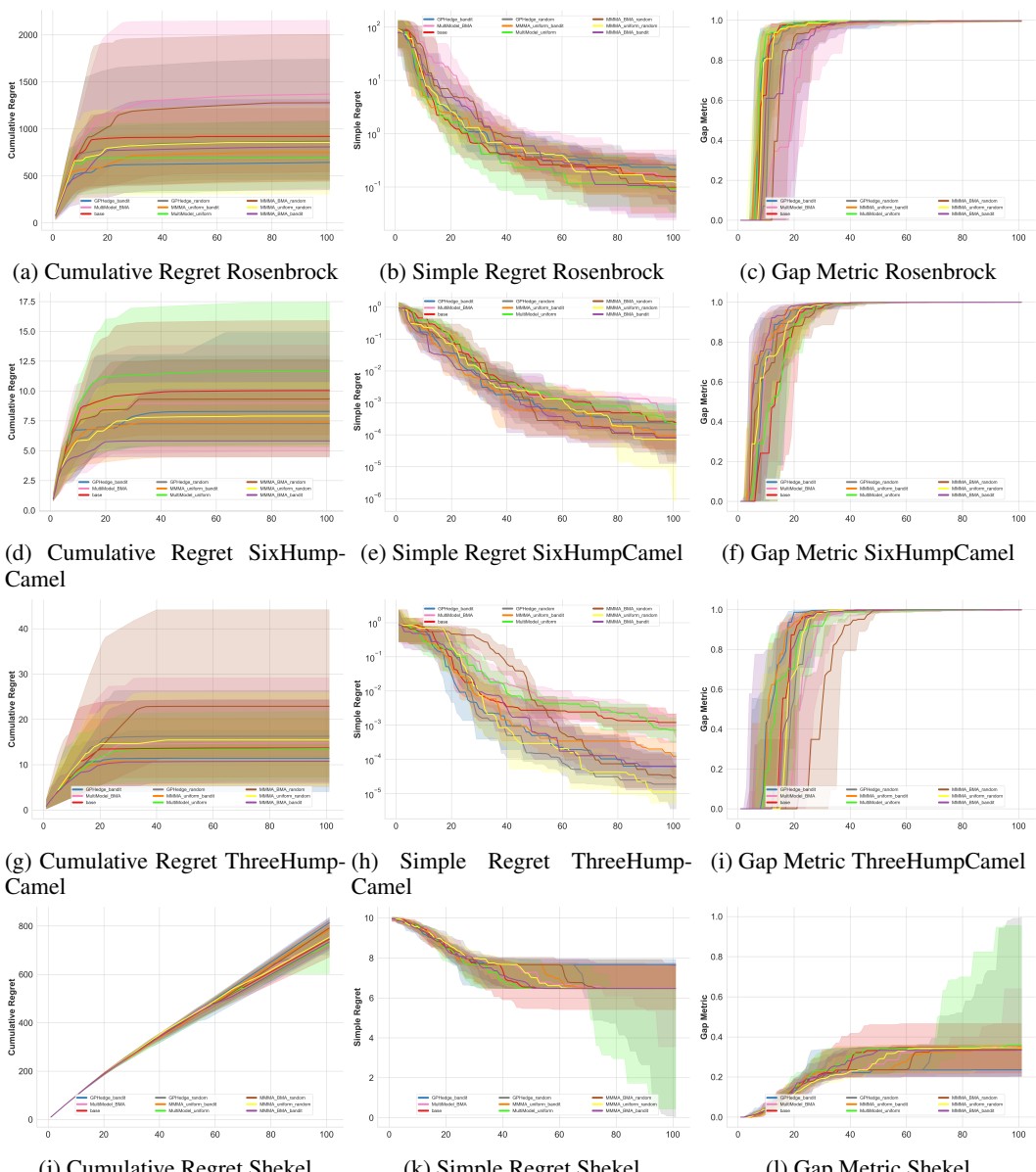

Figure 8: Cumulative Regret, Simple Regret and Gap metric

