# OpenReview forum: "Ensemble Mashups: A Simple Recipe For Better Bayesian Optimization"
_NeurIPS.cc/2024/Workshop/BDU — NeurIPS BDU Workshop 2024 Poster_

### Official Review · Reviewer_49TP · 2024-09-24
**Ensemble Mashups: A Simple Recipe For Better Bayesian Optimization**

**Rating:** 7
**Confidence:** 3

**Review:**

## Summary and Contributions

The paper "Ensemble Mashups: A Simple Recipe For Better Bayesian Optimization" introduces MMMA-BO (Multi-Model-Multi-Acquisition Function Bayesian Optimization), a novel framework that ensembles multiple Gaussian Process (GP) kernels and acquisition functions (AFs) to enhance the robustness and performance of Bayesian Optimization (BO). Traditional BO relies on a single kernel and AF, whose selection critically influences optimization outcomes. MMMA-BO addresses the lack of an automatic selection mechanism by sampling pairs of kernels and AFs from predefined pools at each iteration, thereby mitigating the risk of suboptimal choices. The authors conduct extensive experiments on 16 test functions from Botorch's suite, demonstrating that MMMA-BO outperforms single-level ensemble methods in terms of gap metrics, simple regret and cumulative regret. Additionally, the paper provides a thorough analysis of different ensemble strategies, including fixed uniform weights and performance-guided adaptive weights.


## Strengths

1. **Innovative Ensemble Approach**: The integration of both multiple kernels and acquisition functions in a unified ensemble framework is a significant advancement. By considering the interaction between kernels and AFs, MMMA-BO offers a more comprehensive solution compared to previous methods that only ensemble one component.
2. **Comprehensive Experimental Evaluation**: The authors evaluate MMMA-BO across a diverse set of 16 test functions, providing robust empirical evidence of its effectiveness. Running 25 independent simulations for each function adds credibility to the results.
3. **Clear Algorithmic Description**: Algorithm diagrams provides a clear and concise description of the MMMA-BO process, facilitating understanding and potential replication by other researchers.


## Weaknesses

1. **Limited Theoretical Analysis**: While the empirical results are promising, the paper lacks a theoretical foundation explaining why ensembling both kernels and AFs leads to improved performance. Providing theoretical guarantees or insights into the convergence properties of MMMA-BO would strengthen the contribution.
2. **Scalability Concerns**: Although the experiments are thorough, the paper does not address how MMMA-BO scales with an increasing number of kernels and AFs. Discussing computational complexity and scalability would provide a more comprehensive understanding of the method's applicability to high-dimensional or large-scale optimization problems.
3. **Ablation Studies Missing**: To isolate the impact of ensembling kernels versus AFs, ablation studies that systematically evaluate the contribution of each component would be beneficial. This would help in understanding whether the combined ensemble offers synergistic benefits or if most gains are attributable to one of the two components.

---

### Official Review · Reviewer_wu3D · 2024-10-04

**Rating:** 6
**Confidence:** 4

**Review:**

The authors investigate Bayesian optimization where data are acquired using a mixture of acquisition functions applied to a mixture of kernels.

The experimental evaluation in the paper is quite thorough, with many benchmarks. The thoroughness of this evaluation shows that there is no supreme combination of kernels and acquisition functions that works on every problem.

The setup seems a bit unprincipled, especially for Bayesian optimization. A mixture of GPs with different kernels is not itself a Bayesian model average, since the components in a Bayesian model average must themselves be weighted based on a posterior belief on the choice of model. Similarly, acquisition functions are generally derived through a principled formulation of Bayesian decision theory, and a mixture of acquisition functions doesn't have such a formal basis.

The authors also do not elaborate on optimizing an ensemble of acquisition functions. While certain acquisition functions like EI/PI/UCB have simple closed-form expressions amenable to gradient-based optimization, other acquisition functions like KG or Thompson sampling have more complicated procedures for optimization. Can the proposed method handle these other acquisition functions?

With all that said, I think this work could benefit from discussion at the workshop and so I lean towards acceptance.

---

### Decision · Program_Chairs · 2024-10-09

Accept (Poster)